# Neuroinflammation in Parkinson’s Disease: From Gene to Clinic: A Systematic Review

**DOI:** 10.3390/ijms24065792

**Published:** 2023-03-17

**Authors:** Carlos Castillo-Rangel, Gerardo Marin, Karla Aketzalli Hernández-Contreras, Micheel Merari Vichi-Ramírez, Cristofer Zarate-Calderon, Osvaldo Torres-Pineda, Dylan L. Diaz-Chiguer, David De la Mora González, Erick Gómez Apo, Javier Alejandro Teco-Cortes, Flor de María Santos-Paez, María de los Ángeles Coello-Torres, Matías Baldoncini, Gervith Reyes Soto, Gonzalo Emiliano Aranda-Abreu, Luis I. García

**Affiliations:** 1Neurosurgery Department, “Hospital Regional 1° de Octubre”, Institute of Social Security and Services for State Workers (ISSSTE), México City 07300, Mexico; neuro_cast27@yahoo.com (C.C.-R.); dra.dylandiaz@outlook.com (D.L.D.-C.); 2Neural Dynamics and Modulation Lab, Cleveland Clinic, Cleveland, OH 44195, USA; 3Brain Research Institute, Universidad Veracruzana, Xalapa 91192, Mexico; cupae_030315@hotmail.com (K.A.H.-C.); michelvichy@gmail.com or zs19026360@estudiantes.uv.mx (M.M.V.-R.); cristoferjzc@gmail.com (C.Z.-C.); kabuto.najaraya@gmail.com (O.T.-P.); delamora3david@gmail.com (D.D.l.M.G.); flordemariasantez@gmail.com (F.d.M.S.-P.); garanda@uv.mx (G.E.A.-A.); luisgarcia@uv.mx (L.I.G.); 4Pathology Department, “Hospital General de México”, Dr. Eduardo Liceaga, México City 06720, Mexico; erickapo@hotmail.com (E.G.A.); javiertc924@hotmail.com (J.A.T.-C.); 5Research Department, “Instituto Nacional de Ciencias Médicas y Nutrición Salvador Zubiran”, México City 14080, Mexico; angiecoellot@gmail.com; 6Laboratory of Microsurgical Neuroanatomy, Second Chair of Gross Anatomy, University of Buenos Aires, Buenos Aires C1052AAA, Argentina; drbaldoncinimatias@gmail.com; 7National Cancer Institute (INCAN), México City 14080, Mexico; drgervith@gmail.com

**Keywords:** Parkinson’s disease, gene expression, molecular pathways, cellular alterations, neuroanatomical alterations, neurobehavioral, clinical manifestations

## Abstract

Parkinson’s disease is a neurodegenerative disease whose progression and clinical characteristics have a close bidirectional and multilevel relationship with the process of neuroinflammation. In this context, it is necessary to understand the mechanisms involved in this neuroinflammation–PD link. This systematic search was, hereby, conducted with a focus on the four levels where alterations associated with neuroinflammation in PD have been described (genetic, cellular, histopathological and clinical-behavioral) by consulting the PubMed, Google Scholar, Scielo and Redalyc search engines, including clinical studies, review articles, book chapters and case studies. Initially, 585,772 articles were included, and, after applying the inclusion and exclusion criteria, 84 articles were obtained that contained information about the multilevel association of neuroinflammation with alterations in gene, molecular, cellular, tissue and neuroanatomical expression as well as clinical-behavioral manifestations in PD.

## 1. Introduction

Parkinson’s disease (PD) is the second most common neurodegenerative disease worldwide, where 90% of cases are of sporadic origin and only 10% are associated with congenital alterations. The incidence and mortality of PD has increased significantly in recent years, so it represents a major health problem mainly in subjects whose age is between 45 and 70 years old, in whom the incidence is substantially higher [1,2].

The characteristic manifestations that allow identifying a diagnosis of PD are asymmetric and slowly progressive resting tremors; cogwheel rigidity; bradykinesia and loss of postural reflexes; as well as non-motor alterations, which may occur several years before motor manifestations, and may include anosmia, constipation, depression or apathy, sleep disorders and alterations in memory capacity; and later symptoms of autonomic dysfunction, pain, cognitive impairment, language alterations and psychosis [1,3].

The development and progression of PD manifestations have been mainly related to the significant decrease of the population of dopaminergic neurons (DA neurons) in the substantia nigra pars compacta (SNPc), in conjunction with the presence of Lewy bodies (LBs), which are intracytoplasmic inclusions, including insoluble and misfolded α-synuclein (α-syn) aggregates, considering LBs and the presence of α-syn as a pathognomonic molecular characteristic of PD [3,4]. Such alterations are usually more frequent in structures, such as the basal ganglia, the Locus Coeruleus, the raphe nuclei, the thalamus, the amygdala and the cerebellum [1,3,5].

Despite the first description of this disease more than 200 years ago [6], its etiology is still not fully understood; however, the importance of neuroinflammation in the pathophysiology of PD has been recognized [7]. The process of “neuroinflammation” implies the inflammatory mechanisms that occur in the central nervous system (CNS) and involve both the innate and the adaptive immune system, becoming ambivalent, since it promotes the activation of neuroprotective and neurodevelopmental mechanisms, while inducing damage to nervous tissue contributing to the process of neurodegeneration in neurodegenerative diseases, such as PD [8].

To this end, the multilevel study of PD is a useful tool in advancing the understanding of this disease and its relationship with neuroinflammation as well as the design of new treatment strategies [5,9]. That is why the aim of this article is to collect and present the most current advances regarding PD and neuroinflammation from a perspective that integrates multiple levels of study, from alterations in gene, molecular, cellular and tissue expression, to neuroanatomical alterations and clinical manifestations, related to the process of neuroinflammation in PD. The aim and results of this work are summarized in Figure 1.

## 2. Materials and Methods

### 2.1. Objective

The primary aim of this systemic review is to collect and present the advances regarding PD from a perspective that integrates the different levels of study, from alterations in gene, molecular, cellular and tissue expression, to neuroanatomical alterations related to the most recent evidence of the clinical characteristics of PD.

### 2.2. Search Strategies

A systematic electronic search was conducted according to the Preferred Reporting Items for Systematic Reviews and Meta-Analyses (PRISMA) guidelines [10] in the PubMed, Google Scholar, Scielo and Redalyc databases, with a focus on clinical studies, review articles, book chapters and case studies. That search provided insights into neuroinflammation in Parkinson’s disease from a multi-level approach that included alterations in gene, molecular, cellular, tissue, neuroanatomical expression and clinical manifestations. All databases were last consulted in October 2022. Database-specific filters were used as needed to complete searches across all specified databases. Search strategies and keywords are illustrated in Figure 2 and Table 1 (Appendix A).

### 2.3. Selection of Studies

Using inclusion and exclusion criteria, the authors independently examined the titles and summaries of retrieved studies to determine those that required further evaluation. After removing the duplicates, the authors further evaluated the potential studies that were identified through the search strategy. When the relevance was determined, the full texts of the articles were retrieved and evaluated for possible inclusion, based on the relationship of the contents with the neuroinflammation process in Parkinson’s. The study procedure is depicted on a PRISM flowchart as seen in Figure 2.

## 3. Results

The authors selected 84 articles for the analysis (See Appendix A), with a focus on a total of five levels of alterations associated with neuroinflammation in Parkinson’s disease: alterations of genetic and molecular expression, cellular and tissue alterations, pathological anatomical alterations and behavioral alterations. Each subtopic was developed via a synthesized description based on data obtained from clinical studies and review articles.

### 3.1. Genetic and Molecular Alterations of Parkinson’s Disease Associated with Neuroinflammation

In the 1990s, autosomal dominant and recessive inheritance patterns were identified that allowed more than one gene to be considered as being involved in the development of PD. In 1997, the first PD-related gene was identified, and mutations are currently recognized in 23 genes (PARK1 to PARK23) that are directly related to PD as shown in Table 2 and Table 3 and in four genes (POLG, GBA1, TMEM230 and LRP10) whose alterations contribute to its development as shown in Table 4 [11].

It should be noted that only seven genes (SNCA, LRRK2, VPS35, PRKN, PINK1, DJ-1 and GBA) have been categorically associated with PD [12], while the remaining genes have been classified as possible risk factors due to lack of replication and functional validation [13].

#### 3.1.1. Autosomal Dominant PD

##### SNCA Gene

The SNCA/PARK1 gene encodes for the α-syn protein, which is made up of 140 amino acids that are distinguished by three domains: the N-terminal region of amphipathic nature, the non-amyloid component (central region) and the C-terminal region of acidic nature [14]. This protein is located mainly in the presynaptic ends and plays a regulatory role in vesicular traffic; in addition, it participates in functions, such as memory, recognition and dopaminergic neurotransmission [15].

At the gene level, three classes of mutations have been described: nonsense point mutations in the coding region, repeated dinucleotide variation in the promoter region and locus multiplications, including duplications and triplications [16]. The first reported mutation was an A53T change in the N-terminal region of the α-syn; subsequently, more point mutations were identified in the same region, such as A30P, E46K, H50Q, G51D and A53E [17].

The fact that the mutations are in the same domain has suggested a relationship between each domain and the pathogenesis of PD; for example, since the N-terminal domain contains repetitions of the highly conserved motif (KTKEGV) whose orientation is towards the mitochondria, it has been associated with the mitochondrial dysfunction observed in PD. In particular, the A53T mutation has been linked with suppression of autophagy, mitochondrial dysfunction and endoplasmic reticulum stress-mediated cell death pathways. In addition, the A30P variant favors binding to receptors on the lysosomal membrane to inhibit chaperone-mediated autophagy, which would result in the accumulation of α-syn in the LBs [18].

The second domain, called NAC, is a hydrophobic core that has a structure of β sheets that tend to form oligomers. This promotes the aggregation and fibrillation of α-syn intracellularly, which is the main component of LBs [19].

The third, C-terminal domain is acidic in nature, which facilitates protein–protein interactions and promotes the formation of pathogenic fibrils and, thus, the formation of Lewy bodies. In addition, it presents interactions with ligands, such as dopamine (DA) and tropomyosin receptors related to kinase B (TrkB), thereby, inhibiting the signaling pathway of brain-derived neurotrophic factor (BDNF)/TrkB and causing the death of DA neurons [18,20].

Mutations in SNCA (encoding α-syn) are considered to be the major pathogenic genetic factor of cytotoxicity-mediated PD, which includes mitochondrial dysfunction, endoplasmic reticulum (ER) stress, loss of proteostasis, synaptic impairment, apoptosis and neuroinflammation [21].

##### LRRK2 (Leucine-Rich Repeat Kinase 2)

The LRRK2/PARK8 gene encodes leucine-rich repeat kinase 2 (LRRK2), also called dadarin, which exhibits enzymatic activities, such as GTPase and kinase [22]. It is mainly located in the cytoplasm and neuroanatomically in the olfactory bulb, striatum, cortex, hippocampus, midbrain, brainstem and cerebellum, especially in pyramidal, medium and microglia spiny neurons [23,24].

LRRK2 is a multi-domain protein composed of seven domains: the arm region (armadillo repeat); ank domain (ankyrin-like repeat); the LRR domain (leucine-rich repeats) located at the N-terminus; the central GTPase Ras-of-Complex (ROC) domain; COR domain (C-terminus); the kinase domain (KIN) surrounded by protein–protein interaction regions, such as the WD40 domain at the C-terminus; and the LRR domain at the N-terminus [25].

Seven pathogenic PD variables have been identified as distributed in two domains, KIN and ROCO, made up of the ROC and COR domains, which are critical for enzymatic activity. G2019S, I2020T and I2012T are presented in the KIN variables, while R1441C, R1441G, R1441H and Y1699C are presented in ROCO. Although the protein may have other amino acid substitutions, they have not been determined to be pathogenic [24].

In particular, the G2019S mutation promotes α-syn mobility and accumulation through RAB35 phosphorylation, in addition, RAB35 hyperphosphorylation, alters endosomal trafficking and lysosomal degradation in the brains of PD patients. On the other hand, overexpression of α-syn increased LRRK2 kinase activity accompanied by dopaminergic neurodegeneration, while inhibition of LRRK2 attenuated α-syn toxicity in a mouse PD model suggesting bi-directional synergistic regulation between α-syn and LRRK2 [24]. 

This mutation also affects the internalization of the dopamine D1 receptor, negatively affecting the transduction of dopaminergic signals through synaptic vesicle trafficking. In addition, the LRRK2 mutation causes increased kinase activity, thereby, increasing mitochondrial fragmentation and decreasing mitochondrial dynamin-like protein 1 (DLP1)-mediated respiratory chain complex IV activity, thus, resulting in increased ROS production [16,18].

##### VPS35

The gene encoding the vacuolar protein-associated protein (VPS35) is a central component of the retomeric complex that mediates the retrograde delivery of molecules from the endosomes to the Golgi trans-network (TGN) and cell surface and that also binds to MAPL, a mitochondria-associated protein ligase [26,27]. 

The D686N variant (formerly D620N) prevents VPS35 interaction with MAPL, leading to increased mitochondrial fission and fragmentation affecting mitochondrial complex I assembly and activity [27]. In addition to this effect, the D686N variant prevents binding with multifunctional protein 2 that interacts with the aminoacyl tRNA synthetase complex (AIMP2) enabling its increase and translocation to the nucleus where it activates PARP1, which, together with the effects of the D686N variant, enable neurodegeneration and neuroinflammation [28].

The R524W variant of VPS35 is relevant in PD through the dysfunction of the retromer complex that affects the delivery of cathepsin D (CTSD) to the lysosome. Consequently, the lysosomal proteolysis is altered [16,18], which leads to the exacerbated accumulation of α-syn within DA neurons by the dysfunction of VPS35 [29].

##### GBA1 (Glucosylceramidase Beta 1)

The GBA1 gene encodes the enzyme glucosylceramidase 1 (GCase 1), also called β-glucocerebrosidase. This is a membrane-bound lysosomal hydrolase [20] that catalyzes the cleavage of glucosylceramide (GlcCer or also called glucocerebroside) and glucosylsphingosine (GlcSph) into glucose and ceramide or glucose and sphingosine, respectively [30]. To perform its activity, it binds to LIMP-2 (lysosomal integral membrane protein 2) and is transported to the lysosomes, where it binds to the cofactor, saposin C, increasing its catalytic activity, protecting it from proteolysis and directing the substrate to the active center of the enzyme [31].

Between 7% and 20% of patients with PD have a GBA mutation [32]. There were 130 GBA1 mutations reported in patients with PD, and the most frequent variants were L444P, N370S, E326K and T369M. The D409H variant is particularly frequent in the European population, the R496H in Ashkenazi Jews and the H255Q in East Asian populations [33]. Such mutations are associated with decreased GCase activity leading to lysosomal accumulation of GlcCer and glycosylsphingosine (GlcSph) [34].

The N370S mutation is more common in PD, causing retention of GCase in the endoplasmic reticulum, thus, causing enlargement and disorganization of this organelle and fragmentation of the Golgi apparatus. Consequently, mitophagy is hindered contributing to the increased and deregulated generation of ROS [35].

The L444P mutation has been linked to increased oligomeric α-syn aggregates, with the most accepted hypothesis being that changes in glycosphingolipid homeostasis can affect the membrane composition and alter lysosomal function and vesicular transport [36]; in turn, α-syn aggregation appears to have a bidirectional relationship with GCase dysfunction [37].

Another hypothesis points to the instability of misfolded GCase, which cannot exit the Golgi apparatus, since it prevents coupling with the carrier protein LIMP-2 [20]. It is further considered that GCase is inactive due to mutations in the active site and that GCase activity is altered due to a Saposin C defect [31].

#### 3.1.2. Autosomal Recessive PD

##### PRKN (Parkin)

The PRKNS gene encodes the Parkin protein, a cytosolic E3 ubiquitin ligase [22] whose function is to covalently bind activated ubiquitin molecules to target substrates for proteasomal degradation, acting mainly on damaged mitochondria that are activated by the putative kinase 1 induced by PTEN (PINK1). Subsequently, the damaged mitochondria are ubiquitinated by the ubiquitin activating enzyme E1 (UbA1) and the ubiquitin conjugating enzymes E2 (UbCH7), and this is a fundamental part of the mitophagy mechanism through the autophagic-lysomal pathway. Parkin also protects mitochondrial function by regulating mitochondrial biogenesis by degrading the repressor of PGC1α activity—called PARIS—allowing for nuclear translocation of PGC1α and transcriptional activation of mitochondria-associated genes [27]. 

Mutations in the PRKNS gene are considered to be the most frequent cause of autosomal recessive PD [16]. A total of 120 pathogenic mutations associated with PD have been identified, the most frequent being R42P, R46P, K211N, C212Y, C253Y, C289G and C441R. Most patients exhibit exon rearrangements in the heterozygous state and insertion and deletion of one or more exons [18,22].

##### PINK1 (Putative PTEN-Induced Kinase 1)

The putative PTEN-induced kinase 1 (PINK1) gene encodes a mitochondrial protein kinase, which, under physiological conditions, is imported into the inner membrane of mitochondria by the complexes of the translocase of the outer membrane (TOM) and the translocase of the inner membrane (TIM). In the inner membrane, it is cleaved by presenilin-associated rhomboidal protease (PARL) located in the IMM and mitochondrial processing peptidase (MPP) in the mitochondrial matrix. This allows degradation of PINK1 by the ubiquitin-proteasome system, which keeps the levels of said protein regulated [38,39].

Under conditions of mitochondrial damage, PINK1 accumulates on the cytosolic face of the outer membrane of mitochondria (OMM), where it phosphorylates ubiquitin at Ser65 (pS65-Ub), initiating the recruitment of Parkin to the OMM where it will be directly activated through phosphorylation, so that PINK1 acts upstream of Parkin [40,41].

Activation of Parkin contributes to the proteolysis of misfolded proteins, such as voltage-dependent ion channel 1 and mitofusin 2 and contributes to mitophagy [42,43]. As a result, PINK1 alterations are associated with mitochondrial dysfunction, dysfunction of metal and calcium ion trafficking and homeostasis, proapoptotic signaling and neuroinflammation and increased susceptibility to α-syn [31,36,38].

For example, the G411S variant of PINK1 generates a decrease in kinase activity causing the accumulation of dysfunctional mitochondria. Other mutations, such as I368N and Q456X, are more associated with reduced stability in the OMM and low levels of the protein, while the A168P, H271Q, L347P and G309D variants weaken Parkin recruitment to damaged mitochondria [18]. In particular, mutations in PINK1 are the second most common cause of autosomal recessive PD [27].

##### DJ-1 (Daisuke-Junko-1)

The DJ-1 gene has seven exons, and various mutations, such as insertions, deletions and substitutions, have been described [20]. It encodes DJ-1 glycase protein, a ubiquitin ligase belonging to the ThiJ/PfPI molecular chaperone superfamily of 189 amino acids, which are present in the cytoplasm and, to a lesser extent, in the nucleus and mitochondria [22].

Its main function is to regulate reactive oxygen species (ROS), given that, during oxidative stress, it is transferred to the nucleus, the mitochondrial matrix and the intermembrane space [41]. Once in the cell compartment, it autoxidizes into the conformation of cysteine 106 (C106), forming the cysteine-sulfonic acid complex. It acts as a transcriptional coactivator, regulating the activity of several transcription factors (TF) [44]—for example, nuclear factor erythroid-2-like 2 (NFE2L2), splicing factor associated with binding proteins (PSF) and p53, which are capable of initiating the expression of several genes of antioxidant and detoxifying enzymes [16].

Likewise, DJ-1 is associated with molecular chaperones, such as mortaline, to protect cells against stress-induced apoptosis [28]. It is even able to act as one, helping newly translated proteins to fold correctly and, in the case of damaged proteins, help in the delivery to proteasomes. In addition, it has been found to participate in the production process and to regulate RNA [22]. 

DJ-1 regulates the positive transcriptional activity of the VMAT2 gene (vesicular monoamine transporter 2), responsible for transporting cytosolic dopamine to synaptic vesicles. In this way, it helps avoid the toxicity of autooxidized dopamine [16] and promotes the transcriptional activity of PINK1. Even in oxidative stress, it is able to interact with Parkin; therefore, it seems to be related to the mitochondrial function in the PINK1/Parkin pathway [44].

Ten different gene transformations have been described, the most relevant variants being D149A, L166P, M26I and E64D. As expected, such mutations affect protein folding, compromising its antioxidant activity and neuroprotective capacity [22]. Decreased lysosomal activity leading to the accumulation of α-syn [29] and increased mitochondrial ROS levels have also been observed to trigger the accumulation of oxidized dopamine, which results in the death of DA neurons [45].

### 3.2. Cellular and Tissue Alterations of Parkinson’s Disease Associated with Neuroinflammation

The main cellular characteristics of PD are the death of dopaminergic neurons located in the SNpc and gliosis, which involves structural and functional modifications of microglia cells and astrocytes as part of a cellular activation process that is key in the development of neuroinflammation. This is a determining factor in the course of the pathophysiology of PD through processes, such as oxidative stress, mitochondrial dysfunction, loss of proteostasis, altered autophage flow, endoplasmic reticulum stress and alterations in lipid metabolism [46]. This is summarized in Figure 3.

#### 3.2.1. Alterations in Lipid Disposition

The cellular presence of LBs is a pathognomonic characteristic of PD. These are composed of fibrillar α-syn and groups of vesicles, lipid droplets, membranes and mitochondria. Since docosahexaenoic acid (DHA) is an abundant fatty acid in neuronal membranes and promotes the aggregation of α-syn in membrane structures of organelles, such as synaptic vesicles, lysosomes and mitochondria, the morphology of the aggregates depends on the ratio between α-syn and DHA.

#### 3.2.2. Mitochondrial Alterations and Oxidative Stress

The presence of α-syn enables the increase in volume and dysfunction of mitochondria and lysosomes. The association of these organelles with α-syn is influenced by the polyphosphoinositide phosphatase SYNJ1 and group VI calcium-independent phospholipase A2, PLA2G6, which is involved in the control of lipid metabolism and located in these organelles. Lastly, these alterations lead to the overexpression of reactive oxygen species, thereby, promoting glial activation and neuroinflammation in a microenvironment of oxidative stress [47,48,49].

Mitochondrial damage in PD is associated with the expression of damage-associated molecular patterns (DAMP) through extracellular vesicles (EVs) that are recognized by the immune system, thus, activating the STING inflammatory pathways, the NLRP3 inflammasome and NFKB [48]. In the PD model of C57BL/6J mice, intrastriatal infusion of erythropoietin (EPO) demonstrated that the interaction of EPO with the 37 kDa isoform of the EPO receptor significantly improved the outcomes of behavioral testing, in connection with the rescue of dopaminergic markers and decreased neuroinflammation by improving the glycolytic rate and counteracting the redox imbalance in mitochondria [50].

Significantly higher levels of senescence markers in astrocytes have been reported in the MPTP mouse model of PD and are associated with the increased loss of DA neurons and behavioral deficits and linked with the increased cellular production and release of pro-inflammatory cytokines and chemokines. Inhibition of senescence was observed both in vitro and in mice when astragaloside IV was administered, thus, decreasing the loss of DA neurons and behavioral deficits by promoting mitophagy, which directly impacted the decreased production of mitochondrial ROS in astrocytes and, thereby, the oxidative stress [51].

#### 3.2.3. Dopaminergic Neuron Alterations and with Neuromelanin

In PD, the estimated cell loss directly correlates with the percentage of neurons pigmented with neuromelanin that exhibit a higher relative conservation of weakly pigmented neurons compared with strongly neuromelanized neurons. Studies in human brains have shown that α-syn is redistributed to the neuromelanin pigment and becomes trapped in granules in the early stages of PD. Subsequent neuroinflammatory changes, such as activation of the innate and adaptive immune responses are highly localized, although not exclusively, within the areas containing neuromelanin. 

In addition to this, the accumulation of intracellular age-dependent neuromelanin in rodents overexpressing human tyrosinase (hTyr) as a model of PD is associated with dysfunction and neuronal degeneration similar to PD, which has led to the proposal that the neuronal accumulation of neuromelanin is a key factor in the development of PD. Likewise, the release of said molecule into the extracellular space has also been associated with potentiation of the neuroinflammatory response through microglia activation [52,53].

In addition to the death of DA cells, morphological alterations have been described in the medium spiny neurons of GABAergic projections of the substantia nigra (SN) (Figure 4). These alterations consist of significant length reduction of the dendritic tree and density of the dendritic column in the advanced-stage PD model of mice that are induced with dopamine depletion through the administration of 6-OHDA.

The reversal of said morphological alterations has been observed at the level of indirect pathway projections expressing D2-like receptors, and they are projected to the external portion of the Globus Pallidus (GPe). This reversal was achieved by prolonged treatment with L-DOPA, suggesting that the depletion of the dopamine level is a crucial element in the development of morphological alterations in the GABAergic projections of the indirect pathway, while structural alterations in the GABAergic spinous projections of the direct pathway are due to the side effects of DA depletion, such as network dynamics and other mechanisms, including neuroinflammation [54].

Damage of DA neurons is considered to be the element that triggers alterations in SN cells, which are derived from the disturbances at the subcellular level, such as mitochondrial dysfunction, defective protein clearance and α-syn release. These alterations involve the production and release of molecules that promote the progressive inflammatory response in the microenvironment of the surrounding brain tissue, thus, affecting other types of neuronal (such as GABAergic neurons) and non-neuronal cells through the neuroinflammatory process by generating a loop that implies a bidirectional synergistic relationship between dopamine depletion and the progression of neuroinflammation [48].

It has been proposed that the absence of channel-dependent L-type dihydropyridine-sensitive dendritic Ca^2+^ waves is responsible for the selective cellular vulnerability in ventral tegmental area (VTA) DA neurons. This is because the accumulation of Ca^2+^, in conjunction with the considerable amount of DA release sites characterizing these DA neurons, is related to the development of metabolic stress primarily associated with increased mitochondrial oxidative phosphorylation to produce the ATP necessary for active membrane transport—given that a biochemical machinery of DAergic SNpc neurons sustained during firing activity can cause a constant production of mitochondrial ROS. This production is an important event in the promotion of neuroinflammation through glial activation in response to the metabolic stress of SN DA neurons [55].

In particular, the death of the DA neurons containing α-syn fibrils promotes the process of nueroinflammation since the α-syn released after cell death is internalized to the microglia cells by phagocytosis and induces the TLR2-NFkB signaling pathway, thereby, promoting the transcription of factors related to proinflammatory and proapoptotic pathways. 

This activation process of the microglia is called microgliosis, and it is characterized by the morphological changes of the microglial cells. The soma goes from having an oval shape surrounded by short extensions to an amoeboid shape with a decrease or loss of cellular processes and an increased release of inflammatory cytokines, which directly affects the survival of the DA neurons and the surrounding cells. Thus, a bidirectional synergistic relationship between microgliosis and the death of DA neurons has been proposed, as it remains uncertain which of the two phenomena is the triggering factor [48].

During the microgliosis process, factor NCX1 is released in the striatum, which acts as a chemotactic that promotes the migration of astrocytes that are activated, and, from this point, they can migrate to the midbrain, increasing the extent of damage present in the main dopaminergic structures affected in PD [56].

#### 3.2.4. Glial Activation

Despite the damage evidenced by microgliosis, it was proposed that this process, as part of the neuroinflammatory response, also promotes a “double rescue” mechanism that consists of increasing the number of intercellular microglial connections, mainly from nan tunneling nanotubes (TNT) and communicating junctions, allowing for the efficient transfer of α-syn aggregates from donor cells to acceptor cells to increase the elimination of said molecule. In addition, this mechanism allows the donation of mitochondria from healthy microglial cells to reduce the inflammatory profile and cell death of dysfunctional microglia containing α-syn fibrils [48].

Transport through TNT has also been reported between astrocytes and between astrocytes and neurons. Through these, the transfer of molecules of the major histocompatibility complex type II (MHC-II) has been observed, thus, suggesting that inflammatory molecules could spread from one astrocyte to another [57]. Evidence regarding this transport mechanism in conjunction with the accumulation of α-syn observed in PD patients has led to the hypothesis that α-syn may have prion-like activity, further adding that α-syn fibril strains may underlie differences in the cellular and regional distribution of aggregates in different synucleinopathies [58]. 

This hypothesis may even be grouped with the one on α-syn misfolding, which tends to occur first in the peripheral autonomic nervous system (particularly in the axonal terminals of the submucosal plexus and the neurons of the myenteric plexus) and, subsequently, migrates to the nigrostriatal areas of the CNS in a prion-like fashion, from glial cell to glial cell invading neurons in its path [59].

#### 3.2.5. Endothelial Cell Activation

As for the participation of endothelial cells in the process of neuroinflammation in PD, high levels of soluble VCAM1 have been observed in plasma of patients with PD. Endothelial cell activation suggests a mechanism that favors the infiltration of immune cells, such as CD4+ and CD8+ lymphocytes, into the brain, whereas, in addition to these findings, the disruption of the blood–brain barrier at the level of the striatum and the midbrain further supports the position that endothelial activation in response to neuroinflammation contributes to the progression of PD [60].

#### 3.2.6. Adaptive Immune Cell Response

When analyzing the postmortem brain tissue of subjects with PD regarding the adaptive immune responses involved in the course of neuroinflammation in PD, an increase in the amount of CD3+ T cells has been observed in the vicinity of neuromelanin + neurons at the midbrain level in a 1:2 lymphocyte-neuromelanin + neuron ratio [61]. The characteristics of monocytes and macrophages identified at lesion locations in PD describe increased release of pro-inflammatory cytokines, accumulation of α-syn and decreased ability to phagocytose [62].

It is presumed that there is crosstalk between T cells and astrocytes in the brain with PD based on the increased expression of MHC-II collocated with the astrocyte marker GLAST, which confirms the increased presence of phosphorylated α-syn in said cells. These cells also surround and establish contact with perivascular and infiltrated CD4+ T cells through the astrocytic expression of the co-stimulatory molecules CD80, CD86 and CD40, which are crucial for the activation of CD4+ T helper cells. This leads to the idea that astrocytes have the ability to act as antigen-presenting cells in the brain of subjects with PD as part of the immunomodulation mechanisms [57].

Active mast cells release mast cell protease, which activates microglia through PAR-2. Glial cells, neurons and mast cells communicate with each other through various signaling pathways, including CD40L, CD40, toll-like receptor 2 (TLR2), TLR4, PAR-2, chemokine receptor 4 (CXC motif) (CXCR4)/CXCL12 and C5a receptor to promote glial cell migration and activation associated with the release of inflammatory mediators. In particular, glial cells and brain mast cells mediate neurotoxic and neurotrophic effects. 

Inflammatory mediators released from the glia in the SN can recruit and activate mast cells, which will enhance the neuroinflammatory response, further promoting the activation of glial cells in a kind of loop promoting cell activation whose constant is to promote neuroinflammation. This serves to explain why mast cells are located next to glial cells in neuroinflammatory conditions of the brain, such as PD [63].

It was proposed that intestinal dysbiosis can cause deregulated activation of the Toll-like receptor (TLR) family, which is expressed in the plasma membrane of intestinal epithelial cells, myeloid cells, T cells, neurons and glial cells. This deregulated activation of TLR initially affects intestinal epithelial cells through the persistent production of pro-inflammatory cytokines and alteration of the intestinal epithelial barrier [48,64].

Observation of atrophic or pycnotic neurons both in the myenteric and submucosal plexus of patients with PD reinforces this assumption. It was further suggested that the pathogen-associated molecular pattern (PAMP), derived from microorganisms causing dysbiosis, can leak to the blood–brain barrier (BBB), cross it and, in conjunction with inflammatory mediators, reach the brain, thereby, promoting the process of neuroinflammation by activating the immune response of the resident glia in conjunction with the adaptive immune cells, which, when presented chronically, enables and increases the disruption of the BBB. This process highlights the relevance of understanding the functioning of the gut–brain axis—particularly the products of microbial metabolism, such as short-chain fatty acids (SCFAs) and tryptophan metabolites that impact neuronal activity and neuroinflammation [48,64,65].

### 3.3. Anatomopathological Alterations of Parkinson’s Disease Associated with Neuroinflammation

#### 3.3.1. Gliosis

Anatomopathological evaluation has made it possible to associate the process of neuroinflammation as a trigger for various crucial alterations in PD from events, such as excessive and irregular microglial activation in the initial phases of PD to the consequent release of proinflammatory cytokines, IL, ROS, apoptosis activation and the loss of DA neurons not exclusive to the SN but also from areas, such as the putamen, the transentorhinal cortex, the cingulate cortex, the temporal cortex and the hippocampus [66,67,68].

In post-mortem studies, cell loss and astrogliosis have been found not only in the SN but also in regions of the subthalamic nucleus (STN), in which the increase in inflammatory markers is associated with greater neuronal loss and astrogliosis in addition to being closely related to the presence of α-syn (predominant in neurites, glia and in multiple LBs) [69].

#### 3.3.2. Pathology through α-syn

In particular, α-syn pathology has been found in six regions of the brain in addition to the STN and the SN (Figure 4). These are the hippocampus, the amygdala and the entorhinal, occipitotemporal, prefrontal and posterior parietal cortices, where, additionally, elevated microglial activation was observed as well as the infiltration of T lymphocytes. Damage was sustained in these areas due to neuroinflammatory processes, and, in areas, such as the frontal cortex and the SN, the amount of proinflammatory cytokine IL-1β was elevated, and this was also present in the amygdala, hippocampus and frontal cortex. These alterations are associated with symptoms such as cognitive impairment. 

The extent of α-syn pathology has been observed from the early stages of Lewy cell pathology in areas, such as the dorsal motor nucleus of the vagal nerve, which is located in the spinal bulb and is the origin of the nerve fibers that innervate the intestine and other visceral organs, as well as in the structures of the olfactory nerve, mainly in the olfactory bulb where a great loss of gray matter has been found, extending to the anterior olfactory tract and nucleus [3,70,71].

#### 3.3.3. Death of DA Neurons

The loss SNpc, Locus Coeruleus (LC) and SN cells is considered to be a product of neuroinflammation, specifically from the increase in cellular exposure to IFNγ, which causes the increased expression of major histocompatibility complex (MHC), which is relevant in the efficient presentation of neoantigens to CD4+ T cells and is associated with the exacerbated activation of microglia [72]. Elevated levels of IL-1β, IL-6 and TNFα in the striatum have been associated with increased apoptosis of DA neurons; although it has been observed that the degree of damage is influenced by sex, since the stereological analyses of the distribution of dopamine receptors (D1) and (D2) are different, and the distribution is greater in men, particularly in the dorsal and ventral portions [60,73]. 

An example of the influence of sexual dimorphism on the effects of neuroinflammation in PD has been observed in the basal ganglia; however, in other characteristically non-dimorphic structures, such as the cerebellum, evidence has been observed on the difference in the bioavailability of levodopa, which is associated with patterns of mitochondrial molecular damage and shows less damage by cytokines and oxylippins in women than in men [74].

#### 3.3.4. Infectious Factors

Despite the fact that neuroinflammation is a determining process in PD and other neurodegenerative diseases, the mechanism of origin of this association remains unknown. In this regard, one of the proposed mechanisms suggests the presence of systemic and local microbial infection of the CNS, based on indicator observations of the presence of microorganisms belonging to the genera Botrytis, Candida, Fusarium and Malassezia in various regions of the brain other than the SNpc, such as the midbrain, hypothalamus, bulge, corpus callosum, caudate and lenticular region and at the peripheral level in the marrow in patients with PD.

Thus, it is considered that these can share polymicrobial infections in the CNS, triggering the immune and neuroinflammatory response and explaining the different motor symptoms of the disease through structural damage [75]. Adding to this assumption, a study of lipid metabolism in PD found that, through a lipidomic profile, it was possible to identify different glycerolipids, saturated fatty acids, primary fatty amines, glycerophospholipids and sterol lipids in the cerebrospinal fluid (CSF) and that these alterations were associated with the promotion of neuroinflammation through the oxidative stress derived from damage to the membrane of organelles (such as the mitochondria) whose membranes are rich in DHA—a fatty acid that promotes the aggregation of α-syn [47,48,76].

#### 3.3.5. Peripheral Nervous System (PNS) Alterations

Significant neuronal damage associated with neuroinflammation indicators, including inflammatory cascades that are associated with colitis syndrome, has been observed in the peripheral nervous system (PNS) of PD patients. This has allowed us to understand why some PD patients present persistent intestinal inflammation compared to healthy controls.

To elaborate, the explanation of this process is based on the vagus nerve connection that innervates the enteric nerve plexus with the CNS. The elevated proteins with angiogenesis and elevated cytokines, such as IL-1α, IL-1β and IL-8, in stool samples in PD patients further reinforces this assumption [72,77]. Another alteration derived from the neuroinflammation in PD observed in SNP occurs in the form of lipidopathy in epidermal cells of PD patients’ skin with GBA mutations [77].

Lastly, pathological alterations associated with the process of neuroinflammation in PD can be classified into two large groups, those associated with the gut–brain axis and those that are limited to neurons located in the brain, whereby the latter group is particularly characterized by the compromise of structures with dopaminergic projections (mainly SN, LC and STN), while the gut–brain axis alteration group presents a wider distribution of compromised structures [3,47,48,70,71,72,76]. This can be seen in Figure 4.

### 3.4. Behavioral Alterations of Parkinson’s Disease and Lifestyles Associated with Neuroinflammation

PD is a disease whose progression comprises multiple exogenous and endogenous factors that favor or delay the development of alterations, such as neuroinflammation, that are crucial in the development and prognosis of PD. In particular, neuroinflammation has been observed in the postmortem brain tissue of patients with PD from the early to late stages, substantiating the occurrence of this alteration in the course of PD, mainly by enabling α-syn aggregation and neurodegeneration. These diseases are seen in Figure 5 [3].

#### 3.4.1. Diet

During the premotor phase of PD, a reduction in neuroinflammation has been suggested, which represents a window of opportunity to prevent the development of the motor symptoms that are characteristic of PD or to delay of the onset thereof, mainly by implementing lifestyles aimed at reducing neuroinflammation, such as a healthy diet with a high intake of fruits, vegetables, seeds and grains, non-fried fish, olive and coconut oil; regular consumption of coffee or tea; exercise and physical activity; and the reduction of chronic stress. 

In contrast, evidence suggests that some factors recognized as harmful to overall health may provide protective effects against the progression of PD when exposure to these is sought in low and controlled doses, such as alcohol consumption or smoking—the latter is mainly due to nicotine stimulus exposure [78,79], although a clinical study that was recently completed did not detect any modification of PD in users of nicotine patches. At the same time, a high consumption of dairy products was associated with an increased risk of PD, attributed to the probable concentration of toxic agents in milk [3].

#### 3.4.2. Physical Activity and Exercise

Exercise and physical activity, even after the first 6 weeks of practice on a 70-min regimen two times per week, improves the quality of life by decreasing motor skill impairment and depression, anxiety and fatigue associated with PD [78,80]. Some disciplines, such as Tai Chi and Yoga, that are low impact and of moderate intensity with emphasis on both skeletal muscle stretching and relaxation, show improvement in motor function and decrease in depression symptoms by increasing the quality of life in patients with PD [81], although the evidence is not conclusive regarding the capacity of exercise to decrease the progression of PD as far as whether the neuroprotective and strengthening properties of the immune system are reinforced [79].

#### 3.4.3. Stress and Comorbidities

Stress can precipitate symptom development of neurodegenerative diseases, including PD [82], which is why PD care strategies have focused on controlling this condition. During the COVID-19 pandemic, patients with PD were reported to have higher levels of stress, depression, anxiety and worse physical activity and quality of life compared to control groups suggesting that there is a two-way synergistic relationship between PD and psychosocial stress [83].

Epidemiological studies have suggested diabetes mellitus as a risk factor for the development and progression of PD [3], in view that the pro-inflammatory state that characterizes diabetes mellitus eventually contributes to the disruption of the blood–brain barrier, allowing for the arrival of peripheral immune cells to the brain tissue, which, together with glycotoxicity, promotes neuroinflammation and, through this, the progression of PD [3,78]. This has led to the proposal of non-pharmacological strategies focused on lifestyles and pharmacological therapies (such as the administration of non-steroidal anti-inflammatory drugs, anti-TNF-α antibody therapy, dextromethrophan and pioglitazone) focused on preventing or delaying the onset and rate of PD progression by decreasing neuroinflammation [78].

#### 3.4.4. Non-Motor Alterations Associated with Neuroinflammation in PD

Patients with PD experience non-motor symptoms related to the accumulation of α-syn and the loss of dopaminergic and noradrenergic innervations in certain brain regions (among which LC stands out) and in some regions of the limbic system, and the main repercussions involve neuropsychiatric or autonomic dysfunctions [78,82,84]. The most frequent are depression [85,86], anxiety [86], fatigue and sleep disorders [82].

##### Depression and Anxiety

The incidence and severity of depression in PD that has been correlated with the administration of dopaminergic drugs [87], in addition to the observations of brain lesion reversal in PD animal models that present characteristics of depression through the administration of dopamine reuptake inhibitors, has led to the assumption that depression in PD is at least partially dopaminergic, although other neurotransmitters, such as acetylcholine, noradrenaline and serotonin, may be involved [88].

Some of the factors that link the development and progression of depression in PD through the promotion of neuroinflammation are considered to be microgliosis in the SNpc and elevated levels of pro-inflammatory cytokines in the brain and blood of PD patients with elevated levels of the soluble receptor interleukin-2 (sIL-2R), tumor necrosis factor alpha (TNFα) and C-reactive protein (CRP) in addition to the dysregulation of genes involved in signaling pathways related to the immune system and lipid and glucose metabolism. Likewise, evidence suggests that depression in PD patients may produce changes in the microbiome through inflammation, thus, reinforcing the growing idea of the significant impact of the gut–brain axis on the development and progression of neurodegenerative diseases, such as PD [85].

Anxiety affects 60% of patients with PD and encompasses generalized anxiety (apprehension, fear and worry), panic attacks and social phobias, and it is commonly, but not always, accompanied by depression. It is most commonly seen in women, patients with disease onset at an early age and in patients with advanced disease, and the fact that it may occur before the onset of the motor signs of PD suggests that this symptom may be related to pathology outside the nigrostriatal pathways [87].

##### Sleep Disorders

Sleep disorders affect most patients with PD, and their prevalence increases with the duration of the disease [87]—one of them being REM sleep behavior disorder (RBD) [86]. This is deemed as a parasomnia characterized by the loss of muscle atony in the REM phase of sleep that results in undesirable motor activity during REM sleep in which people represent or externalize what occurs in their dreams. Additionally, an association has been observed between the presence of RBD in PD and cognitive impairment characterized by decreased attention spans, executive functioning, episodic verbal learning, memory and visuospatial skills [89]. RBD has also been associated with a worse prognosis of depression in patients with early-stage PD, associated with a worse motor and non-motor prognosis and lower life expectancy [85].

Recent evidence suggests that PD may occur in a variety of ways depending on the various endophenotypes with some of them dominated by non-motor symptoms [87], which may be turn out to be the same or even more disabling than the motor ones. They have been shown to have a strong negative impact on patients’ quality of life [85]. In particular, during the COVID-19 pandemic, patients with PD were shown to have higher stress, depression, anxiety, worse physical activity and quality of life compared to control groups [83]. It was also shown that the symptoms perceived to have the greatest negative impact are those that affect social participation, limit mobility or negatively impact the autonomy of patients [79].

##### Cognitive Decline and Dementia

Cognitive decline and dementia are present as a component of late-stage PD, with up to 83% of patients with PD exhibiting some level of cognitive dysfunction. Individuals with a predominantly bradykinetic-rigid form of PD are at higher risk of subsequently developing dementia than people who have a predominantly tremor-prone form of the disease. Late onset dementia is characterized by visuospatial constructive deficits and recognition, semantic and episodic memory loss. These deficits do not originate in the basal ganglia but are associated with Lewy bodies in posterior cortical regions and notably in the parietal and temporal lobes [87].

In particular, early cognitive deficits have been primarily attributed to the loss of DA neurons in the nigrostriatal pathways [84]. Although evidence suggests that PD affects the functions involved in social interaction, there are few conclusive studies researching the realm of social cognition and social decision-making processes in PD [90].

##### Psychosis and Impulsive Behaviors

In late stages of PD, the presence of psychosis has been reported in 40% of cases, where one of the extrinsic causes is the use of dopaminergic agonists [84]. Common symptoms include visual hallucinations and delusions [84,87]. The presence of impulsive behaviors, characterized by the inability to resist inappropriate behavior and a tendency toward rapid, inconsiderate, and uninhibited decisions, have frequently been reported as adverse effects of dopaminergic medication used in the treatment of PD, although the evidence is still not conclusive, are recognized some changes in behavior as shown in Figure 6 [91]. 

Cognitive behavioral therapy is an effective treatment for patients with PD by primarily reducing episodic anxiety, avoidance behavior and social anxiety [92]. Clinical studies have suggested the existence of different phenotypic variants of anxiety and depression in people with PD, ranging from those who exhibit both anxiety and depression to those subtypes with only anxiety or depression [87]. Despite their importance, current treatments for non-motor symptoms are less successful in the long term, and this has a great impact on patients’ quality of life and the burden on caregivers [93].

**Figure 6 ijms-24-05792-f006:**
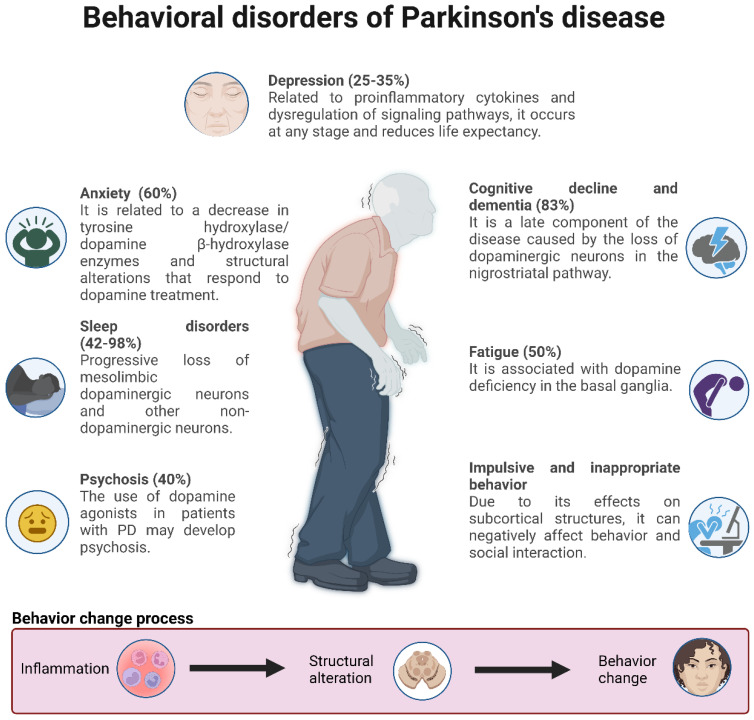
Main behavioral changes in Parkinson’s disease [80,84,85,87,90,93].

## 4. Discussion

The evidence obtained by our systematic search suggests that multilevel alterations associated with the neuroinflammatory process in PD are decisive in the progression of said disease, and therefore they are associated with its characteristics. Despite this, it is clear that it is still necessary to deepen the understanding of the links that associate molecular and cellular mechanisms with clinical behavioral events, and in this framework, it is clear that the scarcity of evidence of this nature is a limitation in understanding PD.

Intracellular alterations associated with neuroinflammation in PD are characterized by the formation of LBs and transfer of α-syn, membranous organelles and various lipids as well as the interaction of α-syn with neuromelanin, allowing for the contrast of the neurotoxic factor of α-syn by oxidative stress against the neuroprotective effect of neuromelanin [52,53,94].

Mitochondrial alterations are also observed, such as the increase in volume, alteration of mitochondrial complexes I and IV, the exacerbated generation of ROS and DAMPs and the decrease in mitophagy, whereby said alterations coincide with the α-syn hypothesis suggested in which the deficiency of mitochondrial complex I in conjunction with toxic 3,4-dihydroxyphenylacetaldehyde (DOPAL) and other elements associated with oxidative stress (such as ROS) plays a relevant role in the cascade of mitochondrial alterations associated with PD [48,50,51,94].

Although the majority of cellular alterations in PD have been described in DA neurons, subsequent DA depletion is associated with a significant reduction in dendritic tree length and the dendritic spine density of SN GABAergic spinous projection neurons, a dysfunctional response has been reported in cholinergic/GABAergic neurons due to increased intracellular chloride resulting in an inefficient inhibitory GABAergic response supplementary to the cholinergic excitatory response derived from poor dopaminergic signaling [48,95].

Thus, the set of cellular alterations in PD affects the glial activation and activation of various cells of the immune system, including endothelial cells that release pro-inflammatory cytokines and chemotactic factors. The role of microglia in the double rescue mechanism was also highlighted as well the hypothesis that astrocytes function as antigen presenters for CD4 T lymphocytes [48,57,58,59,60]. In particular, in patients with PD, circulating CD4 and CD8 T cells infiltrate sites affected by the presence of α-syn and, in response, produce IFNγ and Th1/Th2 cytokines—an event in which the upward expression of MHCII influences myeloid cells of the CNS, thereby, allowing for the recognition of α-syn as a trigger of the immune response [96].

Despite the importance of CD4 T-cell migration for the neuroinflammatory response in PD, it has been observed through in vitro studies that these lymphocytes from PD patients are slower and less viable than those from healthy control groups, mainly due to alterations in mitochondrial positioning and organelle activity—particularly in the study from Mamula et al. They reported alterations in mitochondrial positioning coexisting with a statistically significant decrease in ROS in CD4 T cells from PD patients, which, according to their view, is associated with the poor immune response observed as one of the clinical characteristics of some PD patients. However, the evidence of their in vitro model should be analyzed with caution since this does not rule out the increased production of ROS in brain tissue as an important factor in the development and progression of PD [97,98].

Anatomopathological alterations characterized by astrogliosis mainly include the SN and STN [69], while the regions affected by α-syn depositions are the SN, STN, hippocampus, amygdala, entorhinal, occipitotemporal, prefrontal and posterior parietal cortexes [3,70,71]. In particular, the Braak scale describes the pathological deposition of α-syn in the various brain regions depending on the state of PD progression. This describes the dorsal motor nucleus of the vagus nerve in the medulla and the anterior olfactory nucleus in the olfactory bulb as the regions initially affected; followed by the LC and the SN up to the basal forebrain, the amygdala and the medial temporal lobe structures; and lastly, in late stages, the involvement of cortical areas. However, the generalized deposition of Lewy bodies in both healthy and PD people has led to the alternative assumption of a multicenter origin for PD based on the patterns of neuronal losses [99].

On the other hand, the theory that it originates in certain brain stem structures has been supported by the observation of increased cell loss associated with increased release of inflammatory factors as part of the neuroinflammatory process in the LC [60,72,73]. Specifically, pigment loss in the LC associated with less damage in the rest of the brainstem structures in PD allows for differential diagnoses in regard to parkinsonian syndromes [99].

The presence of infection indicators by microorganisms in the midbrain, hypothalamus, protrusion, corpus callosum, caudate and lenticular nucleus and at the peripheral level in the marrow, as well as the observations of inflammatory cascades that are associated with colitis syndrome in patients with PD, support the relevance of the enteric-brain axis in the genesis process through neuroinflammation and its influence on the development of motor manifestations [72,75].

In regions such as the basal ganglia, the influence of sexual dimorphism modifies the damage associated with neuroinflammation in PD; however, sex affects the bioavailability of dopamine in sexually non-dimorphic structures, such as the cerebellum [74]. The influence of sexual dimorphism and other factors linked to sex are determinants during PD and their response to the various treatment strategies—for example, women present later motor manifestations and a lower propensity to cognitive deterioration and dementia; however, they also present early dyskinesias, postural instability and a greater propensity to neuropsychiatric symptoms, such as anxiety and depression. Although studies evaluating the association between PD and fertile life expectancy, the age of menarche or the age of menopause are not conclusive, the neuroprotective activity of estrogens is recognized [100].

On the other hand, preclinical studies in animal models allow for a more simplified understanding of the pathogenesis of this disease, whether it be symptomatic, morphological or genetic. Through these studies, it has been possible to hypothesize about the complexity of the neural mechanisms underlying motor symptoms, such as tremors and bradykinesia, as well as non-motor symptoms, such as intestinal, muscular and olfactory problems [101,102,103]. Animal models are a great tool in understanding PD, and some of these have given clear indications of the possible brain areas involved beyond the classical theory of indirect and direct pathways, such as the cerebellum and the inferior olive [103].

Similarly, other studies showed that models of genetic defects, such as abnormal α-syn encoding, defects in the autophagy-lysosome system, ubiquitin protease system defects and mitochondria-related dysfunction, can lead to a similar pathogenesis to PD, which is of great relevance in the search for possible primary treatments to combat symptomatology due to genetic defects [103,104,105].

Additionally, clinical studies on PD are rapidly advancing, allowing for not only the observation of symptomatic or pathological implications but also possible treatments that help reduce these alterations [106,107,108]. As the diagnosis of this disease involves multiple factors, the use of other diagnostic tools that fit new technologies is a great candidate for correct interpretation and treatment, including the use of these tools, such as new specific biomarkers (pathological species of α-syn and digital and potential MRI biomarkers), specific treatments for chronic inflammation and the application of machine learning in imaging studies and genetic focuses [106,107,108,109,110,111,112].

The impact of non-pharmacological measures on the development of motor manifestations of PD and the progression of this disease is greater during the premotor phase due to the lower degree of neuroinflammation compared to the rest of the stages and the evidence of a moderate degree of disability that was previously associated only with the late phases—for example, stages I and II of the Hoehn and Yahr scale [76,77,99]. These interventions include a balanced diet, measures to reduce chronic stress and physical activity—the latter stands out due to its neuroprotective properties and strengthening of the immune system [78,79,80,81,113].

The association of chronic diseases as risk factors for the development of PD through the process of neuroinflammation has led to the proposal of pharmacological interventions to prevent and mitigate the development and progression of this disease, mainly focused on anti-inflammatory therapies [3,76]. Primarily among them are the drugs administered in the animal models of PD directed at the transcription factor erythroid factor 2-related nuclear factor 2 (Nrf2)/hemooxygenase-1 (HO-1) pathway, such as Ginalin A, fluprostenol, fucoxanthin, isoorientin [114] and idebenone, which, in cell cultures, demonstrated a transforming effect of the microglia activation state from M1 to M2 and decreased the progress of neurodegeneration in the mouse model of PD by inhibiting the MAPK and NF-κB signaling pathways [115].

Although clinical trials cannot yet be considered conclusive in humans, the glucagon-like peptide 1 (GLP1R) receptor agonist called exenatide improved motor function in patients with PD, and, when analyzing the brain effects of another GLP1R agonist called NLY01, it was suggested that the anti-inflammatory effects of these type of drugs could be mainly responsible for the improvements observed in the clinical trial with exenatide [116].

However, although pharmacological prescription depends on factors, such as age and motor and non-motor symptoms, it is suggested that the complexity of non-motor manifestations is enhanced when dopaminergic agonists are indicated, which are usually used similar to anticholinergics to mitigate the damage associated with the prolonged use of levodopa [91,100].

The neuropsychiatric repercussions associated with the process of neuroinflammation in PD that are part of the wide range of non-motor manifestations include depression [85,86], anxiety [86], fatigue and sleep disorders [82]. The different endophenotypes of PD are decisive in both physical and behavioral characteristics, whereby neuropsychiatric alterations are frequently more disabling than motor alterations [87] given that they can affect social participation, limit mobility and negatively affect the autonomy of patients [79].

It has even been suggested that some targeted therapies against motor manifestations, such as deep-brain-stimulation surgery of the subthalamic nucleus (STN-DBS) could represent a predisposing factor for the development of non-motor alterations, such as depression, impulsivity and decreased verbal fluency [5,117]. Regarding the presence of neuropsychiatric symptoms in PD, these are considered less of a determinant in the development of dementia in comparison with the influence of age and cognitive reserves [118]. Another factor that is profiled as determining the development of dementia in this disease and even in cognitive impairment is the loss of DA neurons from the nigrostrial pathways [84].

Given the above, the complexity of the clinical manifestations and the interactions between the molecular, cellular and physiological pathways leads to challenges in the therapeutic management of patients; therefore, a new perspective that has been developed with the advancement of bioinformatic tools is the generation of polypharmacological strategies, which represent a viable alternative to explore in the future [119,120,121].

## 5. Conclusions

Neuroinflammation occurs in various neurological affections, including neurodegenerative diseases, and, specifically in the case of PD, the characteristics of pathophysiology include genetic, molecular and cellular alterations that converge and maintain cause–effect relationships with each other, which, in turn, promote anatomopathological alterations that ultimately cause motor and non-motor alterations. 

In this sense, alterations in the SNCA, LRK2, VPS35, PRKN, PINK1, DJ1 and GBA genes cause alterations in molecular pathways—for example, the generation of α-syn and its interaction with various cellular elements, such as neuromelanin and mitochondrial membranes, which generates the activation of proinflammatory cascades among which NF-κB, IL-1β, IL-6 and TNFα stand out and cause oxidative stress. This, in turn, impacts the cell biology, as they promote the aggregation of insoluble and misfolded α-syn fibrils forming the LBs, causing mitophagia, autophagy alteration and, finally, neurodegeneration of dopaminergic neurons, microgliosis and astrogliosis.

Together these cause anatomopathological alterations in the SN, ST, hippocampus, amygdala, olfactory nucleus and entorhinal, occipitotemporal, prefrontal and posterior parietal cortexes. In conclusion, these multilevel alterations explain motor manifestations, such as bradykinesia, muscle rigidity, resting tremor and postural instability, and they can produce behavioral disorders, such as fatigue, impulsive and inappropriate behavior, the deterioration of social cognition, depression, anxiety, cognitive impairment, dementia and psychosis.

## Figures and Tables

**Figure 1 ijms-24-05792-f001:**
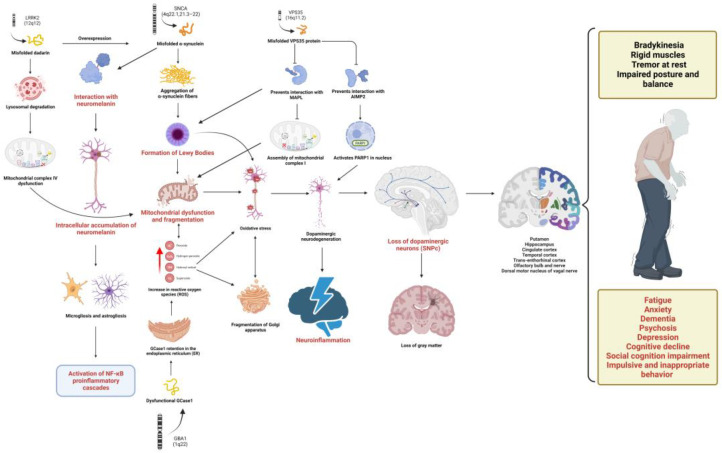
Multilevel alterations in PD pathophysiology. The pathophysiology of PD includes genetic alterations, mainly in the SNCA, LRK2, VPS35 and GBA genes causing the translation of misfolded proteins, including α-syn, dadrina, VPS35 and glycosylceramidase 1, respectively. These dysfunctional proteins converge in various molecular pathways highlighting the activation of proinflammatory cascades and oxidative stress, which present a cause–effect relationship with dysfunctions at the cellular level that involve the accumulation of LBs, mitophagy, utophagy, neurodegeneration of DA neurons, microgliosis and astrogliosis. This leads to anatomopathological changes in the NS, STN, putamen, transentorhinal cortex, cingulate cortex, temporal cortex and hippocampus, the dorsal motor nucleus of the vagal nerve, the olfactory bulb and the olfactory nerve that manifest as motor (bradykinesia, muscle rigidity, resting tremor and postural instability) and non-motor (fatigue, impulsive and inappropriate behavior, social cognition impairment, depression, anxiety, cognitive decline, dementia and psychosis) symptoms in PD. GCase: glycosylceramidase 1, PARP1 (Poly ADP-Ribose Polymerase 1), MAPL (mitochondria-associated protein ligase), AIMP2 (Aminoacyl TRNA Synthetase Complex Interacting Multifunctional Protein 2), NF-κB (Nuclear Factor Kappa B), SNPc (nigra substance pars compacta), SN (nigra substance) and LBs (Lewy Bodies). Created by Biorender.com.

**Figure 2 ijms-24-05792-f002:**
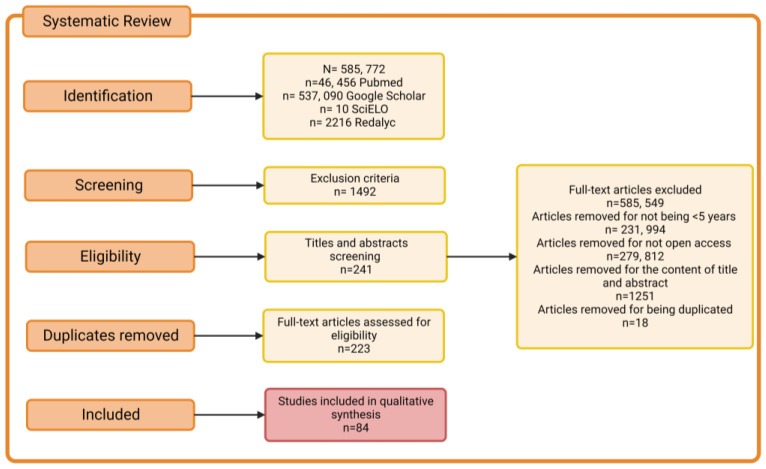
Flow chart of the literature search according to PRISMA guidelines, showing the four search engines used together with the inclusion and exclusion criteria in the order they were applied and the total number of resulting articles.

**Figure 3 ijms-24-05792-f003:**
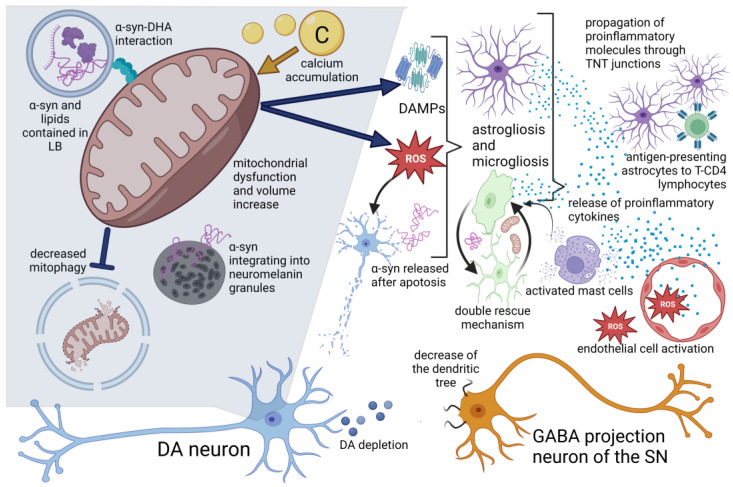
Cellular alterations associated with neuroinflammation in PD. At the intracellular level, the formation of Lewy bodies (LBs) composed of α-synuclein (α-syn), membranous organelles and various lipids stands out. The predilection of α-syn to interact with neuromelanin is observed, being contained in granules during the initial stages of PD. Mitochondrial alterations are also observed, such as increased volume and exacerbated generation of reactive oxygen species (ROS) and the expression of damage-associated molecular patterns (DAMPs), derived from membrane damage by the interaction of α-syn with docosahexaenoic acid (DHA) present in the other mitochondrial membrane and organelles, the intracellular accumulation of calcium and decreased mitophagy. These events participate in dopaminergic neuron (DA neuron) dysfunction and dopamine depletion (DA depletion), which is associated with significantly reduced dendritic arbor length and dendritic spine density of GABAergic spiny projection neurons of the substantia nigra (SN). It is also observed how, after the apoptosis of dysfunctional neurons, α-syn is released, which (together with ROS and DAMP) promotes glial activation and the activation of various cells of the immune system, including endothelial cells that release proinflammatory cytokines and chemotactic factors. This also highlights the role of microglia in the mechanism of double rescue and the hypothesis that astrocytes function as antigen presenters for T-CD4 lymphocytes. Created by Biorender.com.

**Figure 4 ijms-24-05792-f004:**
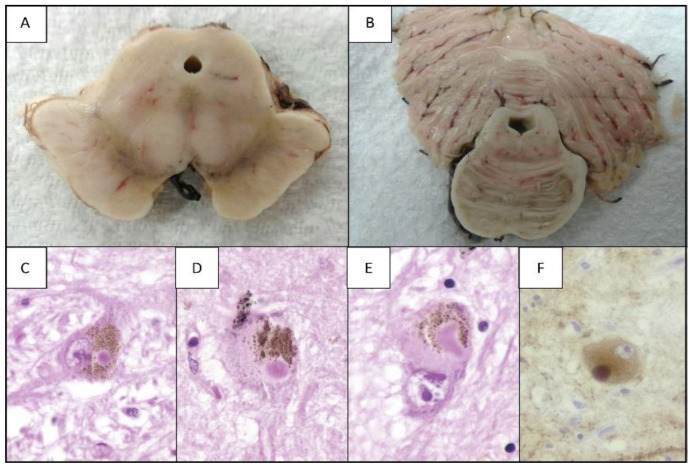
Histopathological features of PD. (**A**) Cross-section of the midbrain, level of the superior cerebellar peduncle: there is depigmentation of the substantia nigra and dilation of the cerebral aqueduct (of Sylvius). (**B**) Cross-section of the pons Varolii: there is slight depigmentation of the locus ceruleus. (**C**–**E**) Hematoxylin/eosin (1000×), microscopic detail of neurons with neuromelanin in the substantia nigra with the presence of a Lewy body. (**F**) Immunohistochemistry with anti-synuclein (1000×), cytological detail of a Lewy body with positive expression.

**Figure 5 ijms-24-05792-f005:**
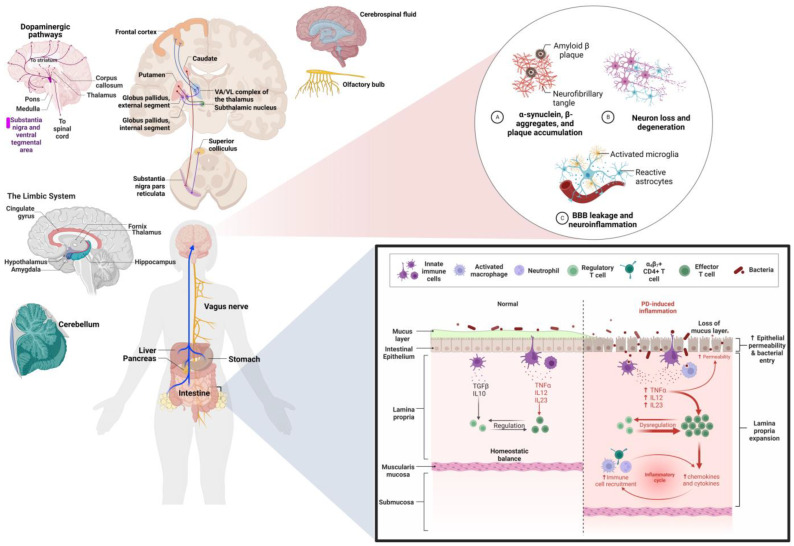
Pathological changes in PD. The diagram indicates the two main pathways (gut–brain and brain neuronal) involved in neuroinflammation associated with PD and shows the main structures in which alterations and damage have been found, both in function and in morphology. These structures extend beyond the SN and are not limited to dopaminergic projections as some of the findings are in the cerebellum, olfactory bulb and cerebrospinal fluid. Created by Biorender.com.

**Table 1 ijms-24-05792-t001:** Search strategies and keywords.

		Inclusion Criteria	Intra-Search Engine Exclusion Criteria					Inter-Search Engine Exclusion Criteria	Total of Articles
Topics to be covered	Search engine	Booleans	Initial articles	<5 years	open access	first 10 pages of results	title and content	no duplicate articles	articles cited in this paper
Genetic and molecular alterations of Parkinson’s Disease associated with neuroinflammation	PubMed	((Molecular basis) OR (genetic basis)) AND (Parkinson’s disease)	1442	474	319	100	37	223	84
	Google Scholar		387,000	267,000	17,000	100	38		
	SciELO		1	0	0	0	0		
	Redalyc		716	117	100	100	4		
Cellular and tissue alterations of Parkinson’s Disease associated with neuroinflammation	PubMed	Parkinson’s disease [title] AND neuroinflammation [all fields] AND tissue alterations [all fields] OR cellular alterations [all fields]	1397	344	201	100	5		
		parkinson’s disease [title] AND AND brain tissue [all fields]	2	1	1	1	1		
		Parkinson’s disease [title] AND neuroinflammation [all fields] AND cell culture [all fields]	35	27	18	18	12		
	Google Scholar	Parkinson’s disease [title] AND neuroinflammation [all fields] AND tissue alterations [all fields] OR cellular alterations [all fields]	17,700	15,200	11,900	100	25		
		Parkinson’s disease [title] AND neuroinflammation [all fields] AND cell culture [all fields]	16,800	15,800	15,600	100	22		
	SciELO	(Parkinson’s disease) AND (neuroinflammation)	6	4	4	4	1		
	Redalyc	Parkinson’s disease AND neuroinflammation AND cells OR tissue	88	36	33	1	1		
Anatomopathological alterations of Parkinson’s Disease associated with neAnatomopathological alterations of Parkinson’s Disease associated with neuroinflammation	PubMed	((Parkinson’s disease) AND (pathology)) AND (neuroinflammation)	1796	1016	654	100	20		
		(((Parkinson’s disease) AND (alterations)) AND (post-mortem)) AND (inflammation)	17	8	7	7	5		
		((((Parkinson’s disease) AND (alterations)) AND (anatomy)) AND (pathology)) AND (inflammation)	243	114	78	78	6		
	Google Scholar	(((Parkinson’s disease) AND (alterations)) AND (post-mortem)) AND (inflammation)	19,900	17,100	100	100	3		
	SciELO	((Parkinson’s disease) AND (pathology)) AND (neuroinflammation)	1	1	1	1	1		
	Redalyc	((Parkinson’s disease) AND (anatomopathological))	318	80	80	80	12		
Behavioral alterations of Parkinson’s Disease and lifestyles associated with neuroinflammation	PubMed	parkinson AND (behavior OR neuroinflammation)	38,486	14,663	8037	100	23		
		parkinson AND (karnofsky OR neuroinflammation) NOT alzheimer	3038	1771	1054	100	8		
	Google Scholar	“parkinson’s disease” AND behavior	91,900	18,100	17,300	100	9		
		“parkinson’s disease” AND “lifestyle changes”	3790	1630	1170	100	3		
	SciELO	parkinson AND behavior	2	2	2	2	0		
	Redalyc	parkinson AND behavior	1094	290	290	100	5		

**Table 2 ijms-24-05792-t002:** Genes associated with autosomal dominant Parkinson’s disease.

Genes Associated with Autosomal Dominant Parkinson’s Disease					
Gene	*locus*	Protein	Genetic Variant	Localization	Pathogenesis in PD
*SNCA*	*PARK1*	α-synuclein	A53T, A30P, E46K, H50Q, G51D and A53E	4q22.1	Oxidative stress, mitochondrial dysfunction and neuroinflammation.
	*PARK4*			4q21.3-22	
*UCHL1*	*PARK5*	Ubiquitin C-terminal hydrolase L1	I93M	4p13-4p14	Uncertain association.
*LRRK2*	*PARK8*	Repetition of leucine-rich kinase 2	G2019S, I2020T, I2012T, R1441C, R1441G, R1441H, Y1699C	12q12	Mitochondrial dysfunction, excitotoxicity, oxidative stress oxidative stress
*GIGYF2*	*PARK11*	GRB10 interacting with GYF2 protein	N457T, N56S, K421R	2q36-37	Uncertain association.
*HTRA2*	*PARK13*	HtrA Serine Peptidase 2	A141S, G399S, R404W	2p13.1	Uncertain association.
*EIF4G1*	*PARK18*	Eukaryotic translation initiation factor 4 Gamma 1	A502V, G686C, R1205H	3q27.1	Defects in the initiation of mRNA translation
*VPS35*	*PARK17*	VPS35 retromer complex component	D686N	16q11.2	Oxidative stress, mitochondrial dysfunction and neuroinflammation.
*DNAJC13*	*PARK21*	DnaJC Heat shock protein family (Hsp40) Member 13	N855S	3q22.1	Dysfunction of endosomal traffic. Decreased exocytosis
*CHCHD2*	*PARK22*	Mitochondrial nuclear retrograde regulator 1	T61I, R145Q	7p11.2	Uncertain association.
*PSAP*	*PARK24*	Prosaposin	C509S	10q22.1	Lysosomal dysfunction

**Table 3 ijms-24-05792-t003:** Genes associated with autosomal recessive Parkinson’s disease.

Genes Associated with Autosomal Recessive Parkinson’s Disease					
Gene	*locus*	Protein	Genetic Variant	Localization	Pathogenesis in PD
*PRKN*	*PARK2*	Parkin	R42P, R46P, K211N, C212Y, C253Y, C289G and C441R	6q25.2-q27	Mitochondrial dysfunction and mitophagy
*PINK1*	*PARK6*	PTEN-induced putative kinase 1	G411S, I368N, Q456X, A168P, H271Q, L347P and G309D	1p36.12	Oxidative stress, mitochondrial dysfunction and neuroinflammation.
*DJ-1*	*PARK7*	DJ-1 Protein	L166P	1p36.23	Oxidative stress, mitochondrial dysfunction Lysosomal dysfunction
*ATP13A2*	*PARK9*	ATPase 13A2	T12M, G533R, F182L, G504R, A746T M810R, G877R	1p36	Mitochondrial dysfunction and mitophagy
*PLA2G6*	*PARK14*	Phospholipase A2 Group VI	D331Y, R635Q, R741Q, R747W	22q13.1	Uncertain association.
*FBXO7*	*PARK15*	F-Box 7 Protein	Not identified	22q12-q13	Uncertain association.
*DNAJC6*	*PARK19*	DnaJ Heat shock protein family (Hsp40) Member C6	Q789X, R927G	1p31.3	Uncertain association Endocytic/lysosomal pathway defect
*SYNJ1*	*PARK20*	Synaptojanin 1	R258Q, R459P	21q22.11	Dysfunction in autophagy
*VPS13C*	*PARK23*	Classification of vacuolar proteins 13 Homolog C	Not identified	15q22.2	Mitochondrial dysfunction

**Table 4 ijms-24-05792-t004:** Genes linked as risk factors for autosomal dominant PD.

Genes Linked as Risk Factors for Autosomal Dominant PD						
*POLG*	DNA polymerase gamma, catalytic subunit	Not identified	Frameshift	15q26.1	Uncertain association.	2004
*GBA1*	Glucosylceramidase β-1	N370S and L444P	Nonsense	1q22	Lysosomal pathway dysfunction Autophagy alteration	2009
*TMEM230*	transmembrane protein 230	R141L	Nonsense	20p13	Oxidative stress, mitochondrial dysfunction and neuroinflammation.	2016
*LRP10*	LDL receptor-related protein 10	Not identified	Nonsense	14q11.2	Uncertain association.	2016

## Data Availability

The review protocol can be accessed at drmarin.neuroscience@gmail.com.

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
