# Peer review of "Neuroinflammation in Parkinson’s Disease: From Gene to Clinic: A Systematic Review"

_ijms, 2023, doi:10.3390/ijms24065792_

Round 1
Reviewer 1 Report
This systemic review's primary goal is to compile and present the latest findings regarding Parkinson's disease (PD) from a perspective that integrates research at various levels, from changes in gene, molecular, cellular, and tissue expression to neuroanatomical changes associated with the most recent clinical evidence of PD's symptoms. The review is exciting, but some concerns need to be addressed. Please see below.
The manuscript did not include search strategies and keywords – it looks like Table 1 is missing.
Several preclinical studies were not included in this review – an example is below:
DOI: 10.3389/fphar.2020.617085
Line 23: define "-EP"
The paragraph, including lines 380 to 392, is very confusing. The data mentioned in this section is poorly reviewed (authors cite only one article under this section, i.e., REF 54). Please elaborate on this section. See also the two comments below:
Line 381: please check the term "median spinous neurons". A more common terminology for these neurons is "medium spiny neurons".
Line 386: please replace "Pale Globe" with "external portion of the Globus Pallidus"
Author Response
1.- Since the text is substantial, we thought putting keywords in the supplementary material would be better. Therefore, table 1 will be in the supplemental material.
2.- We added the preclinical studies as you requested. In addition, we added a picture with anatomopathological elements.
3.- EP is Parkinson´s Disease. We fixed it.
4.- We changed Medium spiny neurons and Pale Globe, and it was reviewed.
Thank you very much for the commentaries we consider very useful.
Best Regards.
Reviewer 2 Report
This review gives a detailed description of Parkinson's disease. Neuroinflammation is one of the pathological features of Parkinson's disease. However, not all pathological events in PD are directly caused by neuroinflammation. I think it is not very appropriate to associate all pathological events in PD with neuroinflammation. The title of this article may be more appropriate to remove neuroinflammation. What is the meaning of neuroinfommation-EP?
Author Response
Our intention is not to inform the lector that Parkinson's disease always is associated with neuroinflammation. Nevertheless, with original articles and other articles, we try to explain the role of neuroinflammation in Parkinson's disease (if there is a role). As you can see in this text, many preclinical studies focus on neuroinflammation. For example: "Contributive Role of TNF-α to L-DOPA-Induced Dyskinesia in a Unilateral 6-OHDA Lesion Model of Parkinson's Disease". This makes me think that neuroinflammation can provoke the symptoms of Parkinson's disease because it creates damage to the DA pathways and, of course, dysfunction in the basal ganglia.
Thank you very much for your commentaries.
Best regards.